# Identifying Mangrove Deforestation Hotspots in South Asia, Southeast Asia and Asia-Pacific

**Samir Gandhi [1,\*] and Trevor Gareth Jones [1,2,3]**

[1]   Blue Ventures Conservation, Level 2 Annex, Omnibus Business Centre, 39-41 North Road, London N7 9DP, UK; trevor@blueventures.org
[2]   Department of Forest Resources Management, Faculty of Forestry, University of British Columbia, 2424 Main Mall, Vancouver, BC V6T1Z4, Canada
[3]   Terra Spatialists, Suite 800, 1199 West Hastings Street, Vancouver, BC V6E3T5, Canada
[\*]   Correspondence: samir@blueventures.org

**Abstract:** Mangroves inhabit highly productive inter-tidal ecosystems in >120 countries in the tropics and subtropics providing critical goods and services to coastal communities and contributing to global climate change mitigation owing to substantial carbon stocks. Despite their importance, global mangrove distribution continues to decline primarily due to anthropogenic drivers which vary by region/country. South Asia, Southeast Asia and Asia-Pacific contain approximately 46% of the world's mangrove ecosystems, including the most biodiverse mangrove forests. This region also exhibits the highest global rates of mangrove loss. Remotely sensed data provides timely and accurate information on mangrove distribution and dynamics critical for targeting loss hotspots and guiding intervention. This report inventories, describes and compares all known single- and multi-date remotely sensed datasets with regional coverage and provides areal mangrove extents by country. Multi-date datasets were used to estimate dynamics and identify loss hotspots (i.e., countries that exhibit greatest proportional loss). Results indicate Myanmar is the primary mangrove loss hotspot, exhibiting 35% loss from 1975–2005 and 28% between 2000–2014. Rates of loss in Myanmar were four times the global average from 2000–2012. The Philippines is additionally identified as a loss hotspot, with secondary hotspots including Malaysia, Cambodia and Indonesia. This information helps inform and guide mangrove conservation, restoration and managed-use within the region.

**Keywords:** mangroves; dynamics; deforestation; hotspot; South Asia; Southeast Asia; Asia-Pacific

## 1. Introduction

Mangroves are distributed in >120 countries throughout the world [1]. Mangrove ecosystems support high floral and faunal biodiversity and provide a diverse range of goods to coastal communities (e.g., food, fuel, building materials). Mangroves also provide key services such as water filtration, mitigating coastal erosion, and storm protection [2–25]. Mangrove forests often have greater carbon stocks than their terrestrial peers, and are therefore important to global climate change mitigation through $CO_2$ sequestration [14]. Despite their value, global mangrove distribution continues to decline primarily due to anthropogenic activities [26,27]. Annual global mangrove loss is estimated at 1%–2% over the past several decades [7,28–32].

The region encompassing South (i.e., S) Asia, Southeast (i.e., SE) Asia and Asia-Pacific is home to approximately 46% of the world's mangroves [33]. Throughout this region, mangroves are typically highly productive ecosystems, containing the oldest and most biodiverse mangrove forests of the world [30,34,35]. Regional rates of loss are also the highest in the world, primarily due to anthropogenic activities [36,37]. Anthropogenic drivers of loss stem from the underlying processes of

population increase, industrialisation, urbanisation and globalisation, i.e., an increasing demand for commodities [27,31]. Primary drivers include conversion of land to aquaculture, oil palm plantations and rice paddies, coastal development, and over-extraction for woody materials [38–46]. Natural phenomena, e.g., rising ocean temperatures and sea-levels and tropical storms also influence mangrove dynamics [7,11,32,38,44,47–52].

While the overall trend in S Asia, SE Asia and Asia-Pacific is mangrove loss, the disparate nature of anthropogenic drivers has resulted in variable dynamics [31]. Some mangrove areas remain relatively intact, owing to characteristics such as remoteness and inaccessibility, or protected status [53]. Some mangrove areas are even increasing in extent following successful reforestation initiatives and/or natural gain [54,55]. Within this region, sub-regions and countries experiencing relatively substantial loss (i.e., mangrove deforestation hotspots) warrant closer attention and intervention (i.e., loss mitigation activities). Up-to-date and accurate information regarding current and historical mangrove distribution and condition is vital to inform conservation, restoration and managed-use. Such information helps countries in their pursuit of environmental targets, e.g., as set out by Millennium Development Goals or the Ramsar Convention on Wetlands [26,53,56]. Remotely sensed data has been widely used to map mangrove distributions and their dynamics from global to local scales and inform mitigation efforts [15,57]. Remotely sensed data is widely accessible and offers a far cheaper alternative to field-based techniques which are only practical at the local-scale [58]. The ease in which derived information can be updated (given the unrivalled temporal resolution of satellite imaging) presents another advantage, and highlights the utility of remote sensing techniques for large-scale mangrove monitoring projects [15,53,59]. Previous research has inventoried and compared mangrove datasets derived from remotely sensed data [60,61]. Hamilton et al. [60] focused on the Western Hemisphere and Oceana, and on datasets since 2000. Hu et al. [61] conducted an inventory at the global scale for 1990–2016. No studies have simultaneously focused on inventory, comprehensive description and comparison of datasets specifically for S Asia, SE Asia and Asia-Pacific, including all historic datasets.

Here we focus on S Asia, SE Asia and Asia-Pacific, collectively referred to as the region of interest (ROI). This report serves to (1) inventory, describe and compare single- and multi-date geospatial datasets derived from remotely sensed data which provide information about the multi-national and/or national distribution of mangrove ecosystems within the ROI, and (2) use multi-date datasets to extract dynamics and identify a short-list of "hotspots" for mangrove loss based on countries which have exhibited the most proportional loss. The findings inform mangrove conservation, restoration and managed-use initiatives within the ROI through targeting deforestation hotspots in greatest need of intervention.

## 2. Experimental Section

### 2.1. Region of Interest

The Region of Interest (ROI) includes 20 countries (Bangladesh, Brunei Darussalam, Cambodia, Fiji, India, Indonesia, Kiribati, Malaysia, Marshall Islands, Micronesia, Myanmar, Palau, Papua New Guinea, Philippines, Singapore, Solomon Islands, Thailand, Timor-Leste, Vanuatu and Vietnam) and 2 territories (Guam and Northern Mariana Islands) across three major sub-regions: S Asia, SE Asia and the Asia-Pacific (Figure 1). Maldives, Nauru and the French territory of New Caledonia were added due to their inclusion in referenced studies and geographic proximity. The ROI contains approximately 46% of the world's mangrove ecosystems largely due to ideal climatic conditions and extensive coastlines [30,33]. Regionally, mangrove ecosystems are exceptionally biodiverse—approximately 80% of all mangrove species are found within the Indo-Pacific between South India and Northern Australia [62], and SE Asia alone boasts 51 species [30] compared to approximately 10 in Africa or the Americas [63]. Throughout the ROI there exists a strong relationship between local coastal populations and mangrove ecosystems [30].

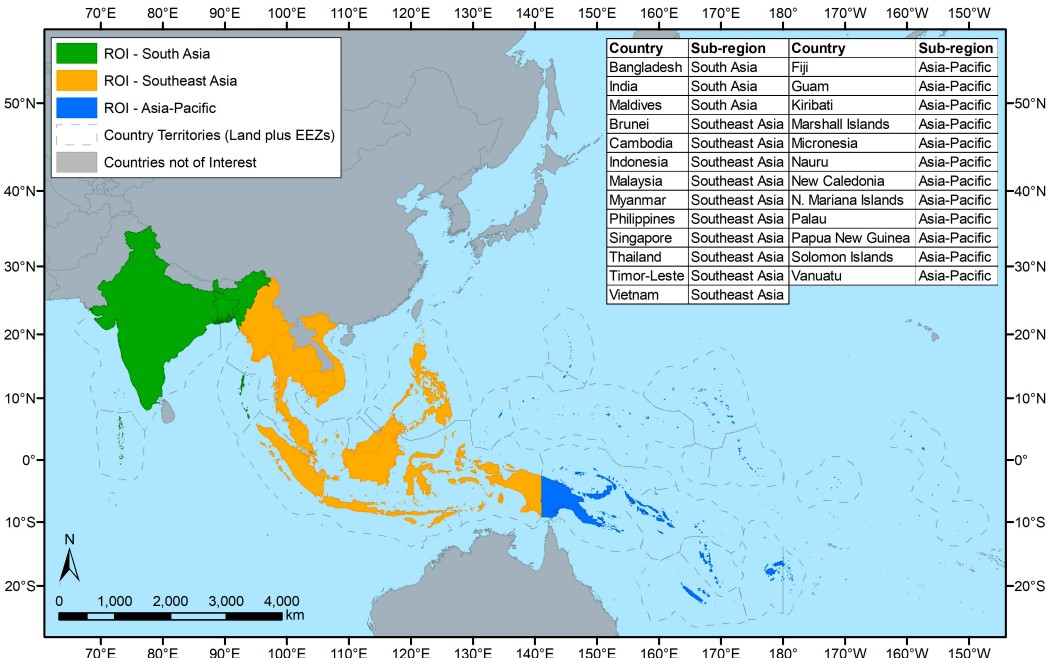

**Figure 1.** The Region of Interest (ROI) covering the three sub-regions of South Asia (green), Southeast Asia (orange) and Asia-Pacific (dark blue). Source: GADM v3.6 [64].

Approximately 5.8% of the world's mangroves are found specifically in the S Asia sub-region [33], distributed primarily in sporadic coastal pockets [19]. This sub-region includes the world's largest mangrove ecosystem, the Sundarbans, covering approximately 1,000,000 ha at the India-Bangladesh interface [65]. Throughout this sub-region, loss is attributed to land-cover conversion, pollution, over-harvesting for timber and natural drivers including cyclones, tsunamis and coastal erosion [66].

SE Asia contains approximately 35.6% of the world's mangroves and notably contains the greatest mangrove diversity representing 51 of the world's known 73 species [30,33,36]. In this sub-region, 30% of all mangrove loss between 2000 and 2012 was attributed to conversion for aquaculture [28,31]. Conversion to rice agriculture has also been a major driver in certain countries, such as Myanmar, whilst in Malaysia and Indonesia loss is mostly attributed to conversion for palm oil plantations [31]. Situated within a highly seismically active zone mangroves here are subjected to tsunamis, as well as other natural loss drivers such as hurricanes and cyclones [62].

Mangroves in the Asia-Pacific make up 4.8% of the world's mangroves distributed across numerous Pacific islands, many of which are volcanic with mountainous terrain which limits low-elevation intertidal areas suitable for mangrove establishment. Mangroves here are typically found in deltas and estuaries of established river systems, the largest of which are in Papua New Guinea, Solomon Islands, New Caledonia and Fiji. Given the low-lying nature of many Pacific islands, climate change, sea-level rise and coastal erosion pose significant threats to the mangroves of this sub-region [62].

*2.2. Inventory, Acquisition and Description of Datasets*

All global, multi-national and national-level mangrove datasets providing single or multi-date coverage within the ROI—up until January, 2019—were inventoried through an exhaustive internet-based search and literature review, and contacting experts with regional experience. Where possible, freely available datasets were obtained from online repositories. For datasets not available through repositories, authors were contacted. Table 1 lists the pertinent datasets (or subsets of datasets) that were not acquired as part of this process, providing author details and the names and status of their corresponding datasets. For those datasets that were unavailable to acquire, inventorying and description was possible by referring to relevant literature.

**Table 1.** Datasets not available from online repositories/acquirable from authors, including those yet to be published.

| Organisation | Website | Applicable Sub-Regions/Countries | Dataset | Availability |
|---|---|---|---|---|
| Aberystwyth University | www.globalmangrovewatch.org | Entire region of interest (ROI) | GMW v2.0 [53] | 2010 global distribution currently downloadable; 1996, 2007–2010 and annually from 2015 not yet published |
| Salisbury University | www.salisbury.edu | Entire ROI | CGMFC-21 v1 [67] | 2000–2014 currently downloadable; 2015 not yet published |
| Mangroves for the Future (MFF) | www.mangrovesforthefuture.org | South (S) Asia; Southeast (SE) Asia | Forthcoming | Not yet published |
| National University Singapore (NUS) | www.nus.edu.sg | SE Asia | 2000–2012 loss [31] | Forthcoming |
| US Geological Survey Earth Resources Observation and Science (EROS) Center | www.usgs.gov/centers/eros | Bangladesh, India | 2000–2012 [19] | Forthcoming |
| Forest Survey of India (FSI) | www.fsi.nic.in | India | State of Forest Map 1987–2017 [55] | Available to purchase |
| National Institute for Environmental Studies | www.nies.go.jp | Myanmar | 2000–2014 change dataset [68] | Forthcoming |
| US Geological Survey Earth Resources Observation and Science (EROS) Center | www.usgs.gov/centers/eros | Philippines | 1990, 2000 and 2010 mangrove distribution maps [69] | Forthcoming |

Once inventoried, all single- and multi-date datasets were described based on the following criteria: brief summary of dataset, spatial and temporal coverage, imagery source(s), mapping methods, resulting land-cover classes, accuracy assessment, mangrove distribution, dataset limitations and details of how dataset was acquired.

### 2.2.1. Single-Date Mangrove Distribution

Single-date mangrove distributions were extracted for each country/territory from all inventoried datasets. Where provided, distribution values were extracted from publications and supporting materials (including for those datasets listed in Table 1 already published). For inventoried datasets for which data were actually available/acquired, distribution values were extracted from the actual datasets based on country/territory boundaries. The borders of countries/territories were geographically defined by combining political boundaries with corresponding exclusive economic zones (EEZ). Country polygons were sourced from the Global Administrative Boundaries database (v3.6, www.gadm.org) [65] and EEZs from Marine Regions (v10 World EEZ, www.marineregions.org) [70]. Country polygons and EEZs were merged and persisting gaps in coverage removed using GIS software to define distinct boundaries for which mangrove extent was calculated.

### 2.2.2. Multi-Date Mangrove Dynamics

Differences between single-date studies were not used to calculate mangrove dynamics, as results would likely be influenced by disparate mapping methodologies and accuracies. This is backed up by findings in [53], in which the authors identify significant (and variable) discrepancies between different global products ([30,33,53]) at regional and sub-country levels. Therefore dynamics were only extracted and reported on from multi-date mangrove distribution datasets i.e., extracted from publications/supporting material, or directly from datasets, for each associated country/territory as defined by combined country and EEZ polygons.

Building on reported accuracies, accuracy was further qualitatively assessed for acquired multi-date datasets through cross-checking against high-spatial resolution satellite imagery viewable in Google Earth Pro (GEP). The qualitative accuracy assessment (i.e., QAA) of multi-date datasets allowed for a more meaningful assessment of mangrove representation, providing a further indicator

of reliability of this study's dynamics assessment. QAA was only possible when datasets were acquired. Table 1 refers to four multi-date datasets that were not available from online repositories nor acquired from authors ([19,31,68,69]), a further dataset not acquired due to associated cost [55], and two more yet to be (wholly or partially) published ([53,67]). Furthermore QAA was not conducted unless datasets explicitly mapped mangrove forest.

When available, multi-date datasets were typically acquired in raster format, and in a range of coordinate systems necessitating several pre-processing steps to enable QAAs. These steps included (1) extracting the mangrove class from the raster layer, (2) converting mangrove class into a vector layer, (3) aggregating vector layer from multipart features to single part feature layer, (4) applying distinctive symbology fit for the GEP interface, and (5) converting symbolised layer into a KML file for bringing into GEP. In addition, the limited processing power of GEP (regardless of computing power) required simplifying large multi-featured datasets, causing GEP to freeze indefinitely, or run with an unmanageably slow response rate. Extensive trial and error identified a maximum number of 40,000 features at which QAA could be reasonably undertaken. Each QAA was based on 100 × 100 km areas of interest (AOIs) divided into 10 × 10 km boxes. Depending on geographic coverage and initial observations of the internal variability of mangrove ecosystems, 1-2 AOIs were used per dataset. Working from NW to SE, every fourth 10 × 10 km box containing mangroves was selected for spot-checking, such that 25% of each AOI was systematically assessed. For each spot-check, mangrove coverage was assessed against GEP imagery as close to the dataset's date as possible. If imagery was not available within five years of the mangrove dataset's temporal focus, QAA was not undertaken. In some instances, part of an image was cloud-covered or low quality, again limiting the ability to conduct QAA (a limitation noted by Estoque et al. [71]). Drawing on canopy-cover definitions described in Jones et al. [72], four mangrove classes were assessed for each spot-check: (1) closed-canopy: tall, mature stands, >60% closed, (2) open-canopy: medium, short or stunted stands, 30–60% closed, (3) sparse/dwarf: short or stunted stands <30 % closed, often found on the margins of mangrove habitats, or in colder/less favourable climates/conditions, (4) fringing/strip: thin, linear stands typically along coastlines or small inward channels. Each class was assessed as either well-, under- or over-represented. The results of QAA and overall suitability (i.e., temporal and spatial coverage) help contextualise the use of multi-date datasets for assessing dynamics.

## 3. Results and Discussion

### 3.1. Inventory and Description of Datasets

Five global datasets (i.e., A–E), four sub-region-wide datasets (i.e., F–I), three multi-national datasets (i.e., J,K,M) and four single-nation datasets (i.e., L,N–P) are inventoried in this study (Table 2). For all datasets, single- and multi-date mangrove distributions are partitioned according to discrete mangrove coverage (i.e., presence versus absence) versus representations of continuous attributes (i.e., canopy cover). As continuous measures are comparatively lower than discrete measures, the data is split by type of measure to avoid masking patterns within the data.

**Table 2.** Inventory of global, multi-national and national mangrove datasets available for S Asia, SE Asia and Asia-Pacific.

| Data-Set | Type | Spatial Coverage within ROI | Temporal Coverage | Author(s) | Discrete/Continuous |
|----------|------|------------------------------|-------------------|-----------|---------------------|
| (A) | Global | Entire ROI | 2000 | Giri et al. [33] | Discrete |
| (B) | Global | Entire ROI | c. 2000 | Spalding et al. [30] | Discrete |
| (C) | Global | Entire ROI | 2000–2017 for loss (annually); 2000, 2012 for gain | Hansen et al. [73] | Continuous |
| (D) | Global | Entire ROI | 2000–2014 (annually) | Hamilton and Casey [67] | Continuous |

**Table 2.** *Cont.*

| Data-Set | Type | Spatial Coverage within ROI | Temporal Coverage | Author(s) | Discrete/Continuous |
|---|---|---|---|---|---|
| (E) | Global | Entire ROI | 2010 | Bunting et al. [53] | Discrete |
| (F) | Multi-national | S and SE Asia | 2000 | Stibig et al. [74] | Discrete |
| (G) | Multi-national | S Asia | 2000, 2012 | Giri et al. [19] | Discrete |
| (H) | Multi-national | SE Asia | 2000, 2012 | Richards and Friess [31] | Continuous |
| (I) | Multi-national | Asia-Pacific | 2000 | Bhattarai and Giri [62] | Discrete |
| (J) | Multi-national | Bangladesh and Myanmar | 1975, 1990, 2000, 2005 | Giri et al. [75] | Discrete |
| (K) | Multi-national | Bangladesh and Myanmar | 1999 | Blasco et al. [76] | Discrete |
| (L) | National | India | 1987–2017 (every 2 years over period) | Forest Survey India [55] | Discrete |
| (M) | Multi-national | Myanmar, Thailand, Cambodia and Vietnam | 2014 | Clark Labs [77] | Discrete |
| (N) | National | Myanmar | 2000, 2014 | Estoque et al. [68] | Discrete |
| (O) | National | Philippines | 1990, 2000, 2010 | Long et al. [69] | Discrete |
| (P) | National | Papua New Guinea | c. 2002 | Shearman et al. [78] | Discrete |

## (A) Mangrove Forests of the World, 2000, Giri et al. (2011) [33]

The most widely-used and referenced global mangrove dataset is Mangrove Forests of the World (MFW) by Giri et al. [33]. Giri et al. [33] was the first comprehensive global assessment of mangrove distribution produced using satellite imagery, providing spatially explicit information at a moderate spatial resolution (i.e., 30 m) for all countries/territories in the ROI circa 2000. The global dataset was produced using approximately 1000 Landsat images (specific sensor information not reported), subset to include areas where mangroves were likely to occur. A hybrid supervised/unsupervised classification approach, using the (Iterative Self-Organizing Data Analysis) ISODATA clustering algorithm, generated 50–150 spectral clusters and four land-cover classes: mangrove, non-mangrove, barren lands and water. The resulting database was evaluated against existing datasets (findings not reported), whilst qualitative validation by local experts employed high spatial resolution QuickBird and IKONOS imagery. Geometric correction reduced root mean square (RMS) error to $\pm 1/2$ pixel. MFW estimated 6,068,096 ha of mangrove within the ROI in 2000. Given (**A**) [33] is a single-date dataset, QAA was not undertaken. Due to the moderate resolution of Landsat data used, small patches of mangrove (<900–2700 $m^2$) were not well captured in the results. Country-level mangrove extent was extracted from both the dataset (vector format from the UN Ocean Data Viewer [79]; raster format from NASA's Socioeconomic Data and Applications Centre (SEDAC) [80]) and from figures reported in Giri et al. [33]. There are discrepancies between mangrove extents as calculated from the downloaded dataset versus figures published for seven of the countries in the ROI. Discrepancies are <5% except in Indonesia where downloaded data presents 13% less mangrove extent, and Malaysia where data presents 10% more (Table 3). As such (and where possible), figures reported in [33] are favoured over those extracted from the data. Country extent figures are presented in Table 4, Figure 2a,b.

**Table 3.** Examples of discrepancies between figures extracted from publications versus from actual available/acquired datasets.

| Country | Sub-Region | Giri et al. [33] | (E) Bunting et al. [53] |
|---|---|---|---|
| Bangladesh | S Asia | 2.2% | 0.0% |
| India | S Asia | 4.9% | 0.0% |
| Indonesia | SE Asia | −13.0% | 0.0% |
| Malaysia | SE Asia | 9.8% | −0.6% |
| Myanmar | SE Asia | 2.5% | 0.0% |
| Philippines | SE Asia | −1.6% | N/A |
| Papua New Guinea | Asia-Pacific | 1.4% | 2.1% |

**(B) World Atlas of Mangroves (WAM) v3, Spalding et al. (2010) [30]**

Spalding et al. [30] incorporated numerous sources including national- and regional-scale mapping to produce a global atlas of mangroves circa 1999–2003. Where gaps in coverage remained, (A) [33] was used to complete coverage. Of interest to this report are the remote sensing-derived national extents, of which two such studies contribute substantially to WAM. Corcoran et al. [81] used Landsat Thematic Mapper (TM) and Enhanced TM (ETM+) scenes for 1999–2001 to produce mangrove distribution maps over Western and Central Africa. This was later replicated by Spalding et al. [30] over Papua New Guinea and Vietnam. The authors employed an unsupervised classification approach, however information regarding methods (i.e., clustering algorithm, land-cover classes) is limited and error metrics are not provided. Cloud-removal pre-processing was not undertaken, resulting in probable inaccuracies. Rosati et al. [82] harnessed the experience of the United Nations Food and Agriculture Organisation's (FAO) Environmental Climate Change and Bioenergy Division by employing Landsat ETM+ scenes to map mangroves across 31 countries (including 12 within the ROI). Mangrove forest was identified and digitised through qualitative interpretation using bespoke software, rather than through image classification techniques. No error metrics or shortcomings were reported however considerable qualitative validation was conducted by experts with detailed field knowledge, with feedback suggesting a high mapping accuracy and a low level of amendments required. Within the ROI, WAM estimated mangrove extents of 1,034,400 ha in S Asia; 5,104,900 ha in SE Asia; and 571,700 ha across what is a more broadly defined as 'Pacific' (Table 4, Figure 2a,b). Given (B) [30] is a single-date dataset, QAA was not undertaken. The complete WAM layer can be downloaded from the UN Ocean Data Viewer [79].

**(C) Global Forest Change database, 2000–2017, Hansen et al. (2013) [73]**

The Global Forest Change (GFC) database employed Landsat ETM+ satellite imagery to produce a global map of percentage tree cover per pixel (i.e., a general 'forest' class) for year 2000 at a spatial resolution of 30 m. A supervised (bagged decision tree) classification was used to identify tree cover and change, making use of parallel processing in Google Earth Engine. The database therefore also calculates annual deforestation from 2000–2017 showing stand-replacement disturbance or complete removal of canopy-cover (i.e., forest cover loss), and the inverse (i.e., forest cover gain) as a twelve year total between 2000 and 2012. A validation exercise used probability-based stratified random sampling. Global accuracy is reported as 99.6% for areas of forest loss or no loss, and 99.7% for forest gain or no gain. Gain was assigned to pixels in which non-forest changed to forest, with tree crown cover densities > 50%. For tropical regions these accuracy figures are 99.5% and 99.7% respectively [73]. Change dynamics for mangrove forest are calculated and reported by (**D**) [67], and extracted for the ROI in this study (Table 5, Figures 3 and 4). Positional accuracy is not reported by the authors. QAA was not undertaken, with priority given to mangrove-focused datasets, however there are notable limitations with this dataset. Firstly, "forest" includes all forests, making no distinction between terrestrial and mangrove. Richards and Friess (H) [31] further cite the inclusion of plantations or semi-natural forests as a limitation. Secondly, forest is defined using a threshold of >5 m wherein lower-stature mangroves are under-represented or completely left out. Testing the GFC over Ambaro-Ambanja Bays (AAB) in NW Madagascar confirmed this limitation—the GFC displayed no mangrove deforestation, wherein multiple studies confirm loss here is extensive [25,67,83]. The complete GFC layer can be downloaded from the GFC's Data Download page [73].

**(D) Continuous Mangrove Forest Cover for 21st Century (CGMFC-21), Hamilton and Casey (2016) [67]**

The CGMFC-21 product builds on (C)'s [73] use of Landsat ETM+ imagery to map global mangrove change from 2000–2014, at a spatial resolution of 30 m. Two global mangrove extent products were produced by masking GFC using (1) (A) [33] and (2) the Terrestrial Ecoregions of the World (TEOW) [84]. TEOW was not compiled using remote sensing methods so is not considered in this study. Areas of forest and annual change within the masked extent were identified from

GFC annually to produce maps of canopy-cover ($m^2$), resulting in continuous (rather than discrete) mangrove coverage. Areas of mangrove change outside of the baseline area (i.e., (A) [33]) are therefore excluded. Accuracy was tested by way of comparison with the only other study reporting similar resolution continuous forest cover available, the USGS National Land Cover Dataset [85] over the United States. Comparison for mangroves in Florida identified a 3.6% disagreement, providing some assurance of accuracy for CGMFC-21. The positional error of the CGMFC-21 product is that of MFW [33], i.e., a published RMS error of $\pm 1/2$ pixel. The product estimated 4,477,769 ha of mangroves in the ROI in 2000, falling to 4,310,725 ha by 2014 (Table 5). This equals a 3.7% decline in extent between 2000–2014, or 0.27% per annum. However, pixels containing just 0.01% forest canopy cover are included as mangrove—this falls well below commonly used minimum canopy-cover definition of 30% for mangrove forest [e.g., [25,67,83]. Initial comparisons with known areas of mangrove loss (e.g., AAB, Madagascar) indicate that due to the aforementioned limitations, mangrove loss is often under-represented. QAA was further conducted for 2014 data over two AOIs in the ROI: North Sulawesi, Indonesia and the Irrawaddy Delta, Myanmar, confirming that low-stature-mangrove forest was under-represented. Dynamics for 19 countries and one territory are presented in and were extracted from (D) [67] (Table 5, Figures 3 and 4). The complete CGMFC-21 layer can be downloaded from its website (http://faculty.salisbury.edu/~{}sehamilton/mangroves/).

**(E) Global Mangrove Watch 2010—a fused optical with SAR approach, Bunting et al. (2018) [53]**

The Japan Aerospace Exploration Agency's (JAXA) Global Mangrove Watch (GMW) initiative has generated a baseline global mangrove extent map for 2010. Phased-Array L-band Synthetic Aperture Radar (PALSAR) data from the Advanced Land Observing Satellite (ALOS) was used due to its global coverage and sensitivity to the physical characteristics of mangrove forest [59]. Due to limitations with L-band SAR (difficulties in distinguishing mangrove from other forms of vegetation at the landward margin), SAR was fused with Landsat TM and ETM+ data resulting in a global map at 25 m spatial resolution. This project is notable for primarily using SAR data in contrast to most other studies that utilise optical datasets, thus exploiting cloud-free, seamless coverage. The methodology involved four stages: (1) extracting a coastal mask, (2) generating a mangrove "habitat" layer where mangrove can plausibly exist, (3) using PALSAR data to generate an initial baseline, and (4) incorporating Landsat composites to refine and eliminate errors in the baseline. The "habitat" layer was defined and generated using five input variables: (1) longitude and latitude, (2) distance to water, (3) surface elevation, (4) distance to an oceanic layer and (5) distance to existing global mangrove classifications (i.e., (A) [33] and (B) [30]). The authors encourage subsequent mapping efforts to make use of this "habitat" layer, though it is yet to be published on the GMW website [53]. Supervised classifications were produced using the Extremely Randomized Trees classifier which defined 500 estimators, generating water, mangrove and terrestrial non-mangrove classes. A stratified random sampling accuracy assessment identified an overall accuracy of 95.3% (based on 53,878 sample points). The authors do not report on the map's positional accuracy. GMW estimated 5,835,322 ha of mangroves within the ROI as of 2010, however no coverage was included for Fiji, Guam, Kiribati and areas east of the antemeridian due to satellite data unavailability (Table 4, Figure 2a,b). QAA was not undertaken given this is a single-date dataset. However, the authors note that due to the moderate resolution of satellite datasets used, fine-scale features were commonly miss-classified, for example aquaculture features, riverine environments, and coastal fringes. The authors suggest a minimum mapping unit of 1 ha for end-user mapping. From early 2019 the GMW initiative plans to provide additional annual maps for years including 1996 using the "map-to-image" method presented by Thomas et al. [86], and using time-series radar imagery e.g., from JERS-1 and ALOS-2 PALSAR-2. GMW 2010 baseline extents for six countries within the ROI are presented in Bunting et al. [53]) and the remaining extents were extracted from data downloaded from the UN Ocean Data Viewer [79] (Table 4, Figure 2a,b). The complete layer can be downloaded from the UN Ocean Data Viewer [79].

**(F) Land-cover map for South and Southeast Asia, Stibig et al. (2007) [74]**

This study used coarse (1 km) resolution imagery from the VEGETATION (VGT) 1 Earth observing sensor aboard SPOT-4 to map land cover classes for S and SE Asia circa 1998–2000. 26 land cover classes were identified using an unsupervised maximum likelihood classification, including "mangrove forest". Class assignment was validated using Landsat TM images, field knowledge and existing land-cover maps. The classification achieved a mapping accuracy of 72% for the dominant classes of 'forest' and 'cropland', though no error metrics were reported specifically for mangrove. Geometric fidelity is reported as within 500 m. The land-cover map identified 4,820,000 ha of mangrove across the ROI. Whilst VGT's coarse spatial resolution of 1 km makes the study useful at a regional scale, the authors acknowledge an evident impact on accuracy at the national level. This is exacerbated by the fragmented, fringing nature of mangrove habitat, resulting in a complete absence of identified mangrove for much of the ROI. For this reason, this study's areal extents are not included. The data is available to download from European Commission's Joint Research Centre website [87].

**(G) Distribution and dynamics of mangrove forests of S Asia, Giri et al. (2015) [19]**

This study employed Landsat ETM+ to assess mangrove cover change between 2000–2012 in Bangladesh, India, Pakistan and Sri Lanka. Three case studies were also assessed in greater spatial and thematic detail: Indus Delta (Pakistan), Goa (India) and Sundarbans (India and Bangladesh). A supervised Classification and Regression Tree (CART) algorithm was employed in Google Earth Engine, generating mangrove, water and 'others' (i.e., combined barren land, agriculture, habitation) land-cover classes. Visual validation by local experts using existing mangrove distribution datasets and high resolution QuickBird and IKONOS imagery helped to improve classification results. Quantitative accuracy assessment was not undertaken, however positional error was successfully reduced to an RMS of less than half a pixel. Mangrove extent for 2000 was estimated as 421,091 ha in Bangladesh and 371,431 ha in India, falling by 2012 by 2.3% (to 411,487 ha) and 7.6% (to 343,065 ha) respectively (Table 4, Figure 2a). Post-classification change analysis identified mangrove dynamics and attributed change to natural or anthropogenic causes. The data was not available/acquired, therefore QAA was not possible. No notable limitations specific to this study are reported by the authors [19].

**(H) Mangrove deforestation in SE Asia, Richards and Friess (2016) [31]**

Building on the methodology adopted by (D) [67], this study assessed the rates and drivers of mangrove deforestation across ten countries in SE Asia: Brunei-Darussalam, Cambodia, Indonesia, Malaysia, The Philippines, Singapore, Thailand and Vietnam from 2000–2012. The methodology cross-referenced deforested pixels from (C) [73] (making use of Landsat ETM+ scenes) within a mask defined by (A) [33], therefore did not account for mangrove gain. Reported figures reflect rates of mangrove loss rather than net mangrove change, which is likely to have reduced areal figures. Deforestation pixels for each year were subtracted from the previous year's total to estimate annual mangrove distribution. A supervised land-use classification method was then used to identify land-use in mangrove deforestation pixels (as identified by masking (C) [73] by (A) [33]). A 100 bootstraps model was used to assess land-use classification accuracy with a median Cohen κ value of 0.62 and a median accuracy of 68%. Six land-use classes were generated including mangrove-regrowth, thereby providing an alternative measure of mangrove gain, and mitigating against the effects of only extracting loss pixels from GFC. As with (D) [67], the positional accuracy is that of (A) [33], i.e., a published RMS error of ±1/2 pixel. Mangrove extent across these countries totalled 4,627,128 ha in 2000, falling to 4,564,371 ha by 2012 (a 1.4% fall over the period, or 0.1% per annum) (Table 5, Figures 3 and 4). Limitations concerning the use of continuous forest cover measures apply here as with (D) [67], however post-processing removed some anomalous deforestation pixels by applying a "clump function" meaning only deforested pixels adjacent to other deforested pixels forming minimum patches of 0.5 ha were retained. The data was not available/acquired therefore QAA was not undertaken.

**(I) Mangrove Assessment in the Pacific, Bhattarai and Giri (2011) [62]**

This study used Landsat ETM+ data to produce a baseline map of mangrove extent across the Pacific circa 2000 including: American Samoa, Fiji, French Polynesia, Guam, Hawaii, Kiribati, Marshall Islands, Micronesia, Nauru, New Caledonia, Northern Mariana Islands, Palau, Papua New Guinea, Samoa, Solomon Islands, Tonga, Tuvalu, Vanuatu and Wallis and Futuna Islands. Classification and validation methods were similar to those outlined in (**A**) [33] and (**G**) [19], generating mangrove, non-mangrove and water land-cover classes, and achieving an overall accuracy of 93%. Mangrove extent within the ROI was estimated to be 569,350 ha with the vast majority in Papua New Guinea (480,121 ha). Given (I) [62] is a single-year dataset, QAA was not undertaken. The authors could not perform atmospheric correction due to the lack of data regarding atmospheric conditions in the Pacific region, thereby potentially introducing error into results. Distributions for nine countries and three territories are presented in and were extracted from (I) [62] (Table 4, Figure 2a,b). The data was not available to download and was not acquired.

**(J) Mangrove dynamics (1975–2005)—tsunami-affected regions of Asia, Giri et al. (2008) [75]**

This study mapped mangrove extent in regions of S and SE Asia affected by the 2004 tsunami. Approximately 700 Landsat images (Multispectral Scanner System (MSS), TM and ETM+) were used to produce four maps for years 1975, 1990, 2000 and 2005 across only tsunami-affected areas of six countries within the ROI: Bangladesh (entire country), India, Indonesia, Malaysia, Myanmar (entire country) and Thailand. Classification and validation methods are similar to those outlined in (**A**) [33], i.e., use of a hybrid supervised/unsupervised classification approach, using the ISODATA clustering algorithm to generate 50 spectral clusters and identify mangrove classes. Additional ground control points were selected to reduce RMS error to $\pm 1/2$ pixel. Quantitative accuracy assessments were not undertaken due to a lack of ground-truth data for historical dates, whilst a visual qualitative accuracy assessment by local experts using high-resolution QuickBird and IKONOS helped to correct significant errors. In 1975 mangrove extent was estimated at 448,073 ha in Bangladesh and 851,452 ha in Myanmar, falling by 2005 to 438,764 ha and 551,361 ha respectively (Table 4, Figure 2a,b, Figure 4). Whilst the reduction in extent over this period was minimal in Bangladesh (2%), the fall in Myanmar was substantial at 35%. Change analysis was conducted using a post-classification technique, which compared classification results from each of the four imaged years. Shortcomings with this approach as noted by (J) [75] include semantic differences in class definitions, positional and classification errors. Mangrove patches smaller than 1 ha were not mapped, likely reducing distribution figures. Maps were acquired from the authors but analysis was hindered due to unresolved dataset issues stemming from original analysis having been conducted >12 years ago. QAA was performed for the 2005 dataset, the most contemporary year of focus. One AOI was selected for each of the six countries studied. GEP imagery was available within two years of the temporal focus in nearly all cases, except Indonesia wherein the timeliest imagery was from 2012. Increasingly variable results were observed as mangrove cover became more open/sparse. Over-representation was identified in places, particularly where mangrove had been converted to palm oil plantations (e.g., in Malaysia), or clear-cut for agriculture (e.g., Malaysia and India). Whilst mangrove was generally well represented in India, Bangladesh, Myanmar and Indonesia, large stands were missing from Thailand and Malaysia. Dynamics for Bangladesh and Myanmar are presented in and were extracted from (J) [75] (Table 4, Figures 2 and 4).

**(K) SPOT 'Quick-look' map for Bangladesh and Myanmar, Blasco et al. (2001) [76]**

The authors presented this study as the first nationwide estimate of Myanmar's mangrove extent, with Bangladesh also mapped. The study referred to 150 'quick-look' scenes from SPOT 1, 2 and 3 HRV sensors, for c. 1999, to produce mangrove classifications. Due to the modest performance of such a method, 'quick-look' scenes were cross-checked against SPOT1 HRV and RESURS genuine scenes from 1999, using a combination of visual interpretation and supervised classification methods (minimum distance and maximum likelihood [88]). Finally, field surveys were conducted across 25 check plots.

The result was eight mangrove classes, expressed in the published findings as "dense", "degraded" and "reforested". The authors do not report on classification accuracy assessment or positional accuracy. Mangrove extent in Bangladesh amounted to 495,000 ha and in Myanmar 690,000 ha (Table 4). Given (K) [76] is a single-year dataset, QAA was not undertaken. The authors acknowledge limitations with use of 'quick look' data due to modest technical performance, and further investigation on their use is required. They also state that classification accuracy could be improved by 10% if NDVI and empirical thresholds were included. This dataset was not available/acquired.

**(L) National-level mapping of India's mangrove forests from 1987 to 2017, Forest Survey of India (2017) [55]**

The Forest Survey of India (FSI) has been using remote sensing techniques to map the country's forest extent (including mangroves) since 1987, on a two-year cycle. The first assessment was undertaken visually at 80 m spatial resolution (minimum mapping unit of 400 ha), using Landsat MSS imagery. FSI have gradually employed more sophisticated remote sensing methods, switching from visual to digital analysis, and utilising imagery from increasingly more modern Landsat and IRS sensors. The 2017 assessment used imagery from the IRS-Resourcesat-2 LISS-III sensor with a spatial resolution of 23.5 m. A hybrid methodology was employed in which unsupervised classification was aided by visual interpretation undertaken by expert analysts with a strong understanding of the local environment—although the authors state mangrove forest was classified separately owing to their distinct tone and texture in imagery, without detailing further. Three mangrove classes were produced: "Very Dense", "Moderately Dense" and "Open", generating a total extent of 492,100 ha for 2017 (up from 404,600 ha in 1987). This equates to a 0.6% increase per annum over this period (Table 4, Figure 2a). The FSI website provides further detail on their mapping efforts, although no error metrics or shortcomings are provided. The datasets were not acquired due to the associated cost, therefore QAA was not undertaken. Maps and data are available to purchase from their website [55].

**(M) Baseline mapping of aquaculture and coastal habitats, Clark Labs (forthcoming) [77]**

Clark Labs (Worcester, USA) [77] produced a baseline land-cover map for Myanmar, Thailand, Cambodia and Vietnam to inventory pond aquaculture and coastal habitats using Landsat data circa 2014. Supervised classification techniques were applied to Landsat OLI scenes to generate 32 land-cover classes, including mangrove. Full methods, error metrics and shortcomings are to be reported in forthcoming publications. Mangrove extents for the four countries were extracted from data downloaded from the Clark Labs website [77], and are as follows: 29,089 ha in Cambodia, 604,057 ha in Myanmar, 259,678 ha in Thailand and 180,784 ha in Vietnam (Table 4, Figure 2a,b). With a pan-sharpened spatial resolution of 15 m, this dataset offers a superior resolution to the other datasets described (most are at 30 m), but is limited in its spatial and temporal coverage within the ROI. Given (M) [77] is a single-year dataset, QAA was not undertaken.

**(N) Mangrove forest maps over Myanmar for years 2000 and 2014, Estoque et al. (2018) [68]**

Estoque et al. [68] used Landsat TM, ETM+ and OLI imagery to produce mangrove forest maps over Myanmar for years 2000 and 2014 to assess change in mangrove ecosystem services. The study employed the ISODATA algorithm to conduct an unsupervised classification, generating 200 spectral clusters to identify mangrove and non-mangrove classes. Visual assessment was undertaken using Google Earth imagery with the help of technical experts from Myanmar's Forest Department. A classification accuracy assessment was based on the collection of 400 mangrove field validation points, plus 400 non-mangrove validation points from Google Earth to produce confusion matrices for map years 2000 and 2014. Overall accuracies were computed as 91% and 97% respectively. The authors do not report on positional accuracy of the maps. The study estimated Myanmar's mangrove distribution to be 666,759 ha in 2000, declining to 475,637 ha in 2014, representing a 28.66% loss over the period [68] (Table 4, Figure 2a). The findings are further broken down by state. No notable shortcomings are

reported by the authors. Due to the recent publication of this paper, the data was not acquired, therefore QAA was not conducted.

**(O) Mapping and monitoring the Philippines' mangrove forests from 1990 to 2010, Long et al. (2014) [69]**

This study assessed Landsat TM and ETM+ imagery to produce mangrove distribution maps across the Philippines for years 1990, 2000 and 2010. Data and findings from a previous study (i.e., Long and Giri [89]) provided a distribution for 2000, based on an unsupervised classification which employed the ISODATA algorithm for 2000, whilst a supervised decision tree classification approach mapped distributions for 1990 and 2010. Three land-cover classes i.e., mangrove, water, terrestrial non-mangrove were generated. High resolution IKONOS and QuickBird data were used for validation purposes, and an accuracy assessment using stratified random sampling indicated a user's accuracy of 99% and overall accuracy of 93%. The authors do not report on positional accuracy of the map. The study reported a nationwide baseline extent of 268,996 ha in 1990, 256,185 ha in 2000, and 240,824 ha in 2010, indicating a downward trend throughout the period, equaling a 10.5% decline (Table 4, Figures 2a and 4). The authors state a need for future studies to address the misclassification of mangrove as water, particularly around small <900 m$^2$ stands. This was cited as the most common classification error in this study. The data was not available/acquired, therefore QAA was not possible.

**(P) Creation of a forest map in Papua New Guinea, c. 2002, Shearman et al. (2009) [78]**

Shearman et al. [78] produced a land-cover map over Papua New Guinea (PNG) for c. 2002, as part of a study to quantify forest conversion and degradation in the country. Use of Landsat ETM+ scenes was supplemented by Landsat TM, SPOT 4 and 5 where adequately cloud-free ETM+ imagery could not be acquired. Due to the strong tendency for cloud-cover in PNG, imagery could not be acquired for a single year. 2002 is however the proposed imaged year, accounting for 62% of images used. Images were segmented into spectral clusters using 'eCognition' software and the resulting polygons classified through expert visual interpretation. This generated nine land-cover classes including mangroves. Validation was undertaken using low-flying aerial photographic surveys, and identified an overall image classification accuracy of 97.7%. The authors report the positional accuracy of imagery enabled change detection over areas spanning 60–100 m. National mangrove extent was estimated at 574,876 ha (Table 4, Figure 2a). However a lack of wall-to-wall coverage from a narrow period is likely to have undermined results, given the heterogeneous nature of forest change across tropical landscapes such as this [78,90]. However, similarly to (C) [73], forest was defined as having a canopy height of >5 m, which is likely to have under-represented lower-stature trees, including mangroves. Given (O) [69] is a single-year dataset, QAA was not undertaken. This dataset was not available/acquired.

*3.2. Comparison of Datasets*

Whilst studies (A), (B), (E), (F), (G), (I), (J), (K), (L), (M), (N), (O) and (P) report mangrove area in discrete terms i.e., presence or no presence, datasets (C), (D), (H) represent mangrove area using a continuous mangrove cover measure, reporting % forest canopy-cover. Measures of continuous cover result in reduced calculated area. In any given pixel, if mangrove is detected, a discrete measure is likely to represent the entire pixel's area as mangrove e.g., 900 m$^2$ out of 900 m$^2$ represented as mangrove. A continuous mangrove cover measure will represent the pixel in terms of % canopy cover, which if 50% would equal 450 m$^2$. This equates to a 450 m$^2$ difference in that single pixel, resulting in significant discrepancy when assessed at the landscape-level. The presence of sparse or degraded mangrove forest accentuates this effect. This explains the difference in areal extent: figures reported by (C) [73] were on average 38% lower than those reported by (A) [33] for the same year of focus (2000). However another reason for this is likely the omission of mangroves <5 m in height, as per (C) [73]. When compared to (H) [31], figures reported by (D) [67] for the year 2012 were on average 31% lower per country. As well as being lower in absolute terms, rates of mangrove loss were on average roughly double those reported by (H) [31]. There remains a broader question on which measure is more suitable

for calculating mangrove distribution. Hamilton et al. [60] propose that discrete, presence versus no presence pixel measures may be inadequate for applications such as establishing mangrove carbon stocks for programmes such as REDD.

Regardless of the methodology adopted, defining mangrove area spatially is challenging as mangroves often co-exist with other similar coastal habitats (e.g., salt marshes and tidal freshwater forests) [30]. Numerous studies (e.g., [15,53,67,68,91,92]) have noted the lack of consistency in how mangroves are defined (e.g., mangrove forest only; mangrove habitat inclusive of water bodies). This reiterates the need to develop robust and standardised, well-reported methods for accurately quantifying mangrove distribution [89,91–93], and is highly likely to be somewhat responsible for the variability in areal estimates. Likewise, with regards to studies (A) [33] and (E) [53], which were intended to provide global baselines of mangrove distribution for 2000 and 2010 respectively, the papers or supporting material have not provided baseline breakdowns for all countries. Study (A) [33] published figures for the 15 most mangrove-rich countries in the world, whilst (E) [53] published mangrove extents for the top 10. In both cases full geospatial data has been made available on the UN Ocean Data Viewer [79], thus figures for all remaining countries are extractable as undertaken in this study (Table 4, Figure 2a,b). However, for both (A) [33] and (E) [53], in the absence of a full methodology explaining how country boundaries were determined, boundaries produced in this study were potentially defined using disparate approaches (e.g., input datasets; projections), resulting in discrepancies as high as 13% (A) [33] between reported values and extracted values (Table 3). This could have negative consequences on decision-making when used to inform intervention. It would be valuable to resource managers and decision makers for future (and existing) studies to present mangrove distributions by region, sub-region, country and country administrative boundaries e.g., provinces/districts—either in-paper or within supplementary materials. Universal use of an agreed and recognised high-resolution boundary dataset e.g., GADM v3.6 [64] could help facilitate this.

Patterns of mangrove extent change can be drawn from the dynamics data. Of the 15 studies identified, only eight (C), (D), (G), (H), (J), (L), (N), (O) involved multi-year distributions/data from which mangrove dynamics could be calculated. Of the 25 countries/territories within the ROI, mangrove extents in Bangladesh, Brunei, Fiji, Papua New Guinea, Solomon Islands, Thailand and Vietnam have remained relatively steady, exhibiting <3% loss over the time periods studied. Loss in countries and territories with mangrove area <10,000 ha (Maldives, Micronesia, New Caledonia, Palau, Singapore, Timor Leste and Vanuatu) was <2%, but only reported on by (H) [31] and/or (D) [67]. Both of these datasets report continuous rather than discrete mangrove distribution. Whilst this is pertinent for country comparisons in absolute terms (due to the comparatively low coverage of continuous coverage figures), in proportionate terms, when measuring % change between two points in time, the impact is likely to be minimal. Therefore whether a dataset reports coverage in discrete or continuous terms is not regarded as being of significance to the reliability of this dynamics assessment. It should also be noted that (H) [31] potentially underestimates mangrove distribution by only accounting for mangrove loss pixels from GFC (C) [73], and not mangrove gain pixels, whilst (D) [67] was found to under-represent low-stature mangrove forest in the QAA exercise. Both datasets also rely heavily on the MFW (A) [33] and GFC (C) [73] datasets in their methodologies, thereby introducing their respective shortcomings into these studies. Dynamics were not available for Guam, Kiribati, Marshall Islands, Nauru and the Northern Mariana Islands. Furthermore, as most studies presenting dynamics have used temporal intervals of ten years or greater, it is difficult to understand intra-interval change. This makes it difficult to develop a clear understanding of the process of mangrove extent change, and as such more regular intervals (such as the annual global updates proposed in study (E) [53]) would assist in building up that understanding [61].

Of all countries in the ROI, Myanmar exhibited the greatest rate of loss in mangrove extent. (J) [75] reported a 35% decrease from 1975–2005, whilst between 2000–2014 (N) [68] reported a 28.7% decline in distribution. The two studies combined represent a time period of approximately 40 years, thereby capturing longer-term change dynamics and trends. Furthermore, the findings from (N) [68] are

backed up by strong results in accuracy assessments, and findings from (J) [75] are considered reliable given the QAA exercise that found mangroves in Myanmar to be generally well represented. Whilst studies based on continuous measures of mangrove forest cover reported a notable drop in the rate of overall deforestation from 2000, rates of mangrove loss of 5.5% from 2000–2012 [31] and 10.2% from 2000–2014 [67] are still comparatively high post-2000 when compared to other countries in the ROI.

In India a 7.6% decline in mangrove distribution was reported from 2000–2012 (G) [19], again comparatively high for the period. However this figure was heavily influenced by the Sundarbans, the largest contiguous mangrove ecosystem in the world, of which 40% is in India. Literature indicates that the Sundarbans remained relatively stable between 1973 and 2000 with a reported loss in areal extent of approximately 1.2% [66]. The Indian Sundarbans represents approximately 50% of India's total mangrove extent, thus it could be assumed that mangrove loss in India's other mangrove habitats was significantly higher than the nationwide loss of 7.6% as reported by (G) [19]. This is backed up by Thomas et al. [36], who categorise Western India as a "hotspot of mangrove change [ . . . ] which should be prioritised for future monitoring" by using multi-temporal radar mosaics as indicators of change. Conversely, Giri et al. (G) [19] also identify continuous mangrove gain between 1973–2011 when reporting on a case-study over Goa, an area of "substantial mangrove extent" within Western India. (L) [55] (a Government organisation responsible for the assessment and monitoring of forest resources) has identified a broadly upward trend in India's mangrove extent since 1987. Their two-year cycle of remote sensing-based forest assessments has identified increases in national mangrove extent for 10 of 14 updates until 2017, representing a 21.6% gain between 1987–2017. Indeed the latest FSI [55] report in 2017 identified an 18,100 ha increase in mangrove extent since 2015's assessment (a gain of 3.8%). (C) [73] noted in their Supporting Information that India reports "forest gains that are not readily observable in time-series satellite imagery, including this analysis". However the FSI provides a comprehensive breakdown of change dynamics in each of the 12 coastal states, and furthermore dynamics for coastal districts within these states. FSI [55] cite widespread plantation efforts (also noted by Jayanthi et al. [54]) and natural regeneration as factors driving notable mangrove extent increases across five states. This upward trend is echoed by Jayanthi et al. [54], who used Landsat TM and ETM+ scenes to assess mangrove change in the ten largest mangrove forests in India between 1989 and 2013. Mangrove extent across the ten forests was estimated as increasing by 13.2%.

Studies including the Philippines reported variable findings. (O) [69] reported a 10.5% loss in areal extent from 1990–2010, with mangrove reported as present or absent. Findings from (O) [69] are considered to be the most reliable indicator for the Philippines given its single-country focus, and strong performance in the study's accuracy assessment. Studies using continuous mangrove cover as a measure reported very little loss, 0.5% from 2000–2012 [31] and 1.49% from 2000–2014 [67]. (O) [69] generated comprehensive nation-wide maps for 1990, 2000 and 2010 and identified a 6% loss from 2000–2010, thereby contradicting studies assessing loss using continuous measures.

Indonesia has by far the greatest areal extent of mangroves (approximately 45% of mangroves found within the ROI, according to (A) [33]). The rate of loss in Indonesia was relatively low, estimated to be 3.86% between 2000–2014 [67] or as low as 0.46% between 2000–2012 [31]. This loss was not nearly as high as in Myanmar, however in absolute terms according to (D) [67] estimates, loss in Indonesia totalled nearly 100,000 ha from 2000–2014—more than a third of Myanmar's total mangrove area in 2014.

The rate of loss in Malaysia and Cambodia was not as pronounced as in Myanmar, but notable nonetheless. In Malaysia, loss was reported at 2.83% from 2000–2012 [31] and 5.58% between 2000 and 2014 [67]. In Cambodia, loss was reported at 2.28% from 2000–2012 [31] and 5.42% from 2000–2014 [67].

There were difficulties associated with acquiring some of the datasets identified pertinent to this study, particularly those from studies published more than five years ago. In some cases data was provided without sufficient (or any) supporting information or metadata, including fundamental characteristics such as projection information. This undermined the process of describing inventoried datasets when information was not acquired or identified, and slowed down or halted efforts

considerably. Online portals such as the UN Ocean Data Viewer [79] are valuable repositories of geospatial datasets pertaining to the management of coastal environments, but no single repository contains all or even most of the inventoried datasets for the ROI. Uploading additional pertinent historic and future datasets, with all accompanying metadata, would greatly support management efforts, and would improve the spatial and temporal coverage, and accuracy of future inventories, descriptions and comparisons of datasets.

*3.3. Identifying Mangrove Loss Hotspots*

Of the three sub-regions in the ROI, SE Asia has exhibited the most mangrove loss. This finding is backed up by a recent study (i.e., Thomas et al. [36]) which employed methods similar to Lucas et al. [94] to identify mangrove change hotspots between 1996–2010. The study cites mangrove areas of SE Asia as having the highest prevalence of anthropogenic activity versus other regions in the world. When considering individual countries across the datasets inventoried and assessed, Myanmar and the Philippines stand out as mangrove loss hotspots. Myanmar experienced 35% loss between 1975 and 2005 [75] and 27.6% loss between 2000 and 2014 [68], eclipsing rates of loss seen anywhere else within the ROI. Between 2000 and 2012, Myanmar experienced a 5.5% loss [31] and from 2000–2014, a 10.2% loss [67]. Hamilton and Casey [67] cite Myanmar as "the current hotspot for mangrove deforestation", referring to their MFW- and TEOW-derived products (see (**D**) to identify a rate of deforestation four times higher than the global average.

The Philippines experienced 10.5% loss from 1990–2010 with relatively consistent rates across the two decades (**O**) [69]. Richard and Friess [31] and Hamilton and Casey [67] also estimated mangrove losses in the Philippines of 0.5% and 1.5% over their respective periods of study—both lower rates of loss attributed to the use of continuous rather than discrete measures.

Secondary hotspot nations are considered to be Malaysia, Cambodia and Indonesia. Within SE Asia, after Myanmar, these were the next three countries with the greatest rates of mangrove loss as reported by Richards and Friess [31], with 2.8%, 2.3% and 1.7% lost respectively between 2000 and 2012. Hamilton and Casey [67] reported a similar pattern, albeit with higher rates of loss: Malaysia lost 5.6%, Cambodia lost 5.4% and Indonesia lost 3.9%.

Dynamics in India warrant further investigation. Whilst India saw a 7.6% loss from 2000–2012 (above the average across the ROI), the loss appears to be lopsidedly occurring in other parts of India given the relatively well-preserved Indian Sundarbans [75]. Furthermore the conflicting findings for India at the national scale (as identified in this study) highlight the need for further and more regular remote sensing-based monitoring, from which reliable change dynamics can be extracted. Due to conflicting information, we cannot conclude that India is a mangrove loss hotspot.

For smaller countries/territories within the ROI, mangrove change dynamics are lacking in the literature. For Guam, Kiribati, Marshall Islands, Nauru and Northern Mariana Islands, no change dynamics are available at all (and therefore excluded from Table 5). For eight others (including Papua New Guinea), only one set of change dynamics are available (from (H) [31]). Some of these countries may also be loss hotspots however in the absence of data this cannot be verified. The 2010 global baseline published as part of the Global Mangrove Watch initiative (E) [53] potentially heralds a new era of annual mangrove distribution updates. From this a reliable monitoring system is proposed by Bunting et al. [53], using a method that focuses on mapping changes away from the baseline rather than using independently classified baselines [86]). It will also rely upon improved sensors (e.g., Landsat OLI, Sentinel-2), which are rapidly accumulating collections. Furthermore, use of imagery from these sensors within cloud-based platforms (e.g., Google Earth Engine) again holds promise [95–97]. Indeed, studies (C) [73] and (G) [19] employed Google Earth Engine. Such solutions could provide much-needed coverage over countries currently lacking dynamics data and future research should be monitored closely.

**Table 4.** (**a**) Mangrove distribution (ha) by South Asian countries, year and study—discrete datasets; (**b**) Mangrove distribution (ha) by Southeast Asian countries, year and study—discrete datasets; (**c**) Mangrove distribution (ha) by Asia-Pacific countries/territories, year and study—discrete datasets.

(a)

| | (J) | (L) | (K) | (B) | (A) | | (G) | (J) | (E) | | (G) | (L) |
|---|---|---|---|---|---|---|---|---|---|---|---|---|
| Study | Giri et al. [75] | FSI [55] | Blasco et al. [76] | Spalding et al. [30] | Giri et al. [33] | | Giri et al. [19] | Giri et al. [75] | Bunting et al. [53] | | Giri et al. [19] | FSI [55] |
| Year of focus | 1975 | 1987 | 1999 | c. 2000 | 2000 | 2000 | 2000 | 2005 | 2010 | 2010 | 2012 | 2017 |
| Extracted from | Paper | Report | Paper | Book/Papers | Paper | Data | Paper | Paper | Paper | Data | Paper | Report |
| Bangladesh | 448,073 | | 495,000 | 495,136 [R] | 436,570 | 446,340 | 421,091 | 438,764 | 416,300 | 416,290 | 411,487 | |
| India | | 404,600 | | 432,592 [R] | 368,276 | 386,243 | 371,431 | | 352,100 | 352,062 | 343,065 | 492,100 |
| Maldives | | | | | | 85 | | | | | | |
| S Asia total | 448,073 | 404,600 | 495,000 | 927,728 | 804,846 | 832,668 | 792,522 | 438,764 | 768,400 | 768,352 | 754,552 | 492,100 |

[R] Spalding et al. (2010) extracted mangrove extent data from Rosati et al. (2008) ([R]) [82].

(b)

| | (J) | (O) | (K) | (B) | (A) | | (N) | (J) | (E) | | (O) | (M) | (N) |
|---|---|---|---|---|---|---|---|---|---|---|---|---|---|
| Study | Giri et al. [75] | Long et al. [69] | Blasco et al. [76] | Spalding et al. [30] | Giri et al. [33] | | Estoque et al. [68] | Giri et al. [75] | Bunting et al. [53] | | Long et al. [69] | Clark Labs [77] | Estoque et al. [68] |
| Year of focus | 1975 | 1990 | 1999 | c. 2000 | 2000 | 2000 | 2000 | 2005 | 2010 | 2010 | 2010 | 2014 | 2014 |
| Extracted from | Paper | Paper | Paper | Book/Papers | Paper | Data | Paper | Paper | Paper | Data | Paper | Data | Paper |
| Brunei | | | | | | 11,450 | | | | 11,594 | | | |
| Cambodia | | | | | | 47,549 | | | | 59,159 | | 29,089 | |
| Indonesia | | | | | 3,112,989 | 2,707,047 | | | 2,689,000 | 2,689,673 | | | |
| Malaysia | | | | 709,730 [R] | 505,386 | 555,092 | | | 520,100 | 516,858 | | | |
| Myanmar | 851,452 | | 690,000 | 502,900 [R] | 494,584 | 507,158 | 666,759 | 551,361 | 501,100 | 501,132 | | 604,057 | 475,637 |
| Philippines | | 268,996 | | 256,482 [R] | 263,137 | 259,037 | | | | 265,138 | 240,824 | | |
| Singapore | | | | 460 [R] | | 579 | | | | 544 | | | |
| Thailand | | | | 248,362 [R] | | 245,437 | | | | 225,783 | | 259,678 | |
| Timor Leste | | | | | | 1018 | | | | 983 | | | |
| Vietnam | | | | 105,608 [C] | | 215,608 | | | | 160,002 | | 180,784 | |
| SE Asia total | 851,452 | 268,996 | 690,000 | 1,823,542 | 4,376,096 | 4,549,975 | 666,759 | 551,361 | 3,710,200 | 4,430,866 | 240,824 | 1,073,608 | 475,637 |

[C/R] Spalding et al. (2010) extracted mangrove extent data from Corcoran et al. (2007) [81] ([C]) or Rosati et al. (2008) ([R]) [82].

**Table 4.** *Cont.*

**(c)**

| | (B) | (A) | | (I) | (P) | (E) | |
|---|---|---|---|---|---|---|---|
| Study | Spalding et al. [30] | Giri et al. [33] | | Bhattarai and Giri [62] | Shearman et al. [78] | Bunting et al. [53] | |
| Year of focus | c. 2000 | 2000 | 2000 | 2000 | c. 2002 | 2010 | 2010 |
| Extracted from | Book/Papers | Paper | Data | Paper | Paper | Paper | Data |
| Fiji | | | 110,144 | | | | 51,168 |
| Guam | 97 [R] | | 32 | 34 | | | |
| Kiribati | | | 18 | 18 | | | |
| Marshall Islands | | | | 2 | | | |
| Micronesia | 8699 [R] | | 9885 | 9900 | | | 8405 |
| Nauru | | | 3 | 4 | | | |
| New Caledonia | | | 24,767 | 25,099 | | | 29,616 |
| Northern Mariana Islands | | | 27 | 28 | | | |
| Palau | 4853 [R] | | 5681 | 5666 | | | 6176 |
| Papua New Guinea | 426,482 [C] | 480,121 | 486,946 | 480,121 | 574,876 | 476,200 | 486,234 |
| Solomon Islands | 60,252 [R] | | 46,585 | 47,100 | | | 52,729 |
| Vanuatu | 2051 [R] | | 1365 | 1378 | | | 1779 |
| Asia-Pacific total | 502,434 | 480,121 | 685,453 | 569,350 | 574,876 | 476,200 | 636,107 |

[C/R] Spalding et al. (2010) extracted mangrove extent data from Corcoran et al. (2007) [81] ([C]) or Rosati et al. (2008) ([R]) [82].

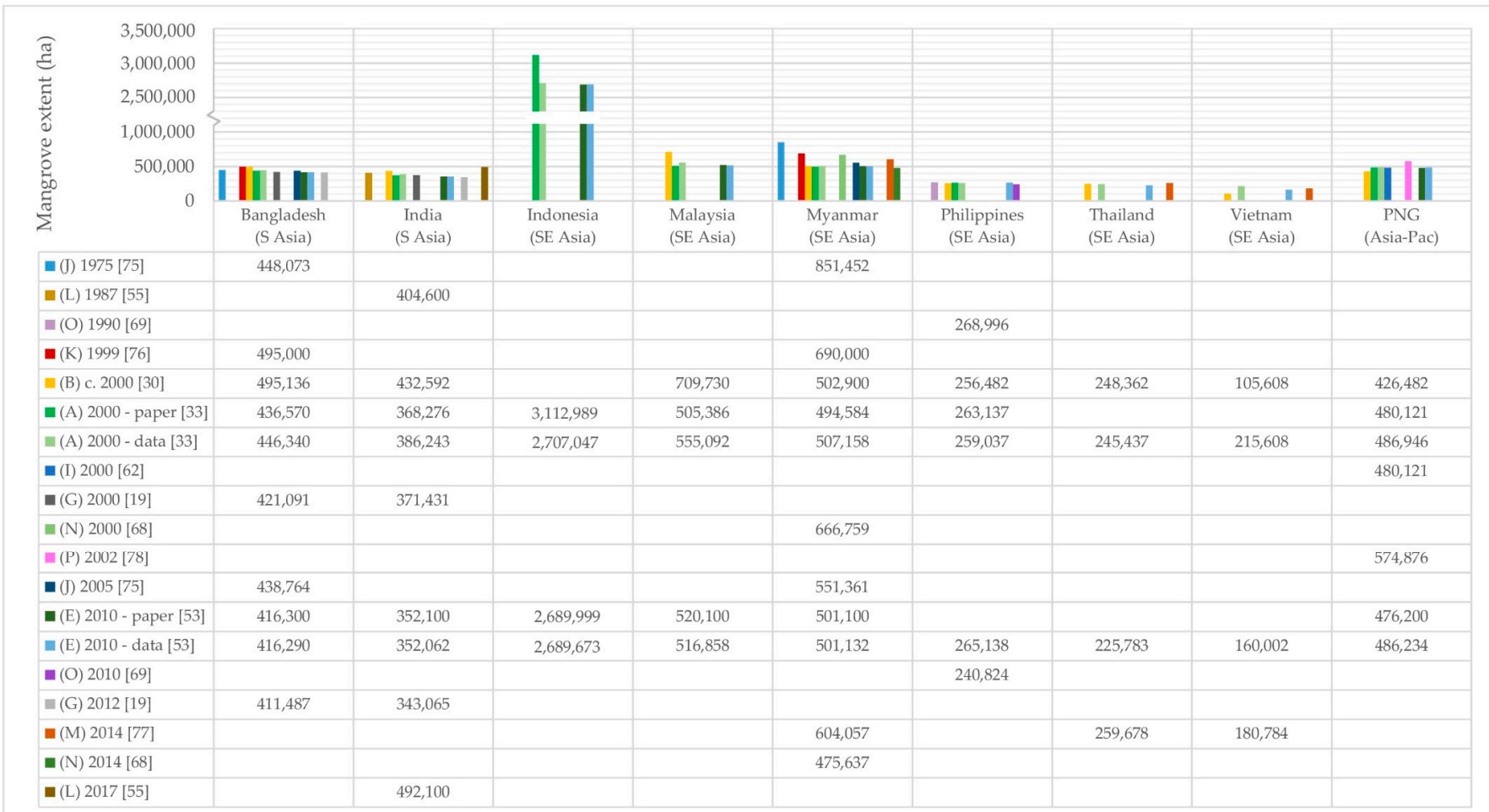

| | Bangladesh (S Asia) | India (S Asia) | Indonesia (SE Asia) | Malaysia (SE Asia) | Myanmar (SE Asia) | Philippines (SE Asia) | Thailand (SE Asia) | Vietnam (SE Asia) | PNG (Asia-Pac) |
|---|---|---|---|---|---|---|---|---|---|
| (J) 1975 [75] | 448,073 | | | | 851,452 | | | | |
| (L) 1987 [55] | | 404,600 | | | | | | | |
| (O) 1990 [69] | | | | | | 268,996 | | | |
| (K) 1999 [76] | 495,000 | | | | 690,000 | | | | |
| (B) c. 2000 [30] | 495,136 | 432,592 | | 709,730 | 502,900 | 256,482 | 248,362 | 105,608 | 426,482 |
| (A) 2000 - paper [33] | 436,570 | 368,276 | 3,112,989 | 505,386 | 494,584 | 263,137 | | | 480,121 |
| (A) 2000 - data [33] | 446,340 | 386,243 | 2,707,047 | 555,092 | 507,158 | 259,037 | 245,437 | 215,608 | 486,946 |
| (I) 2000 [62] | | | | | | | | | 480,121 |
| (G) 2000 [19] | 421,091 | 371,431 | | | | | | | |
| (N) 2000 [68] | | | | | 666,759 | | | | |
| (P) 2002 [78] | | | | | | | | | 574,876 |
| (J) 2005 [75] | 438,764 | | | | 551,361 | | | | |
| (E) 2010 - paper [53] | 416,300 | 352,100 | 2,689,999 | 520,100 | 501,100 | | | | 476,200 |
| (E) 2010 - data [53] | 416,290 | 352,062 | 2,689,673 | 516,858 | 501,132 | 265,138 | 225,783 | 160,002 | 486,234 |
| (O) 2010 [69] | | | | | | 240,824 | | | |
| (G) 2012 [19] | 411,487 | 343,065 | | | | | | | |
| (M) 2014 [77] | | | | | 604,057 | | 259,678 | 180,784 | |
| (N) 2014 [68] | | | | | 475,637 | | | | |
| (L) 2017 [55] | | 492,100 | | | | | | | |

(**a**)

**Figure 2.** *Cont.*

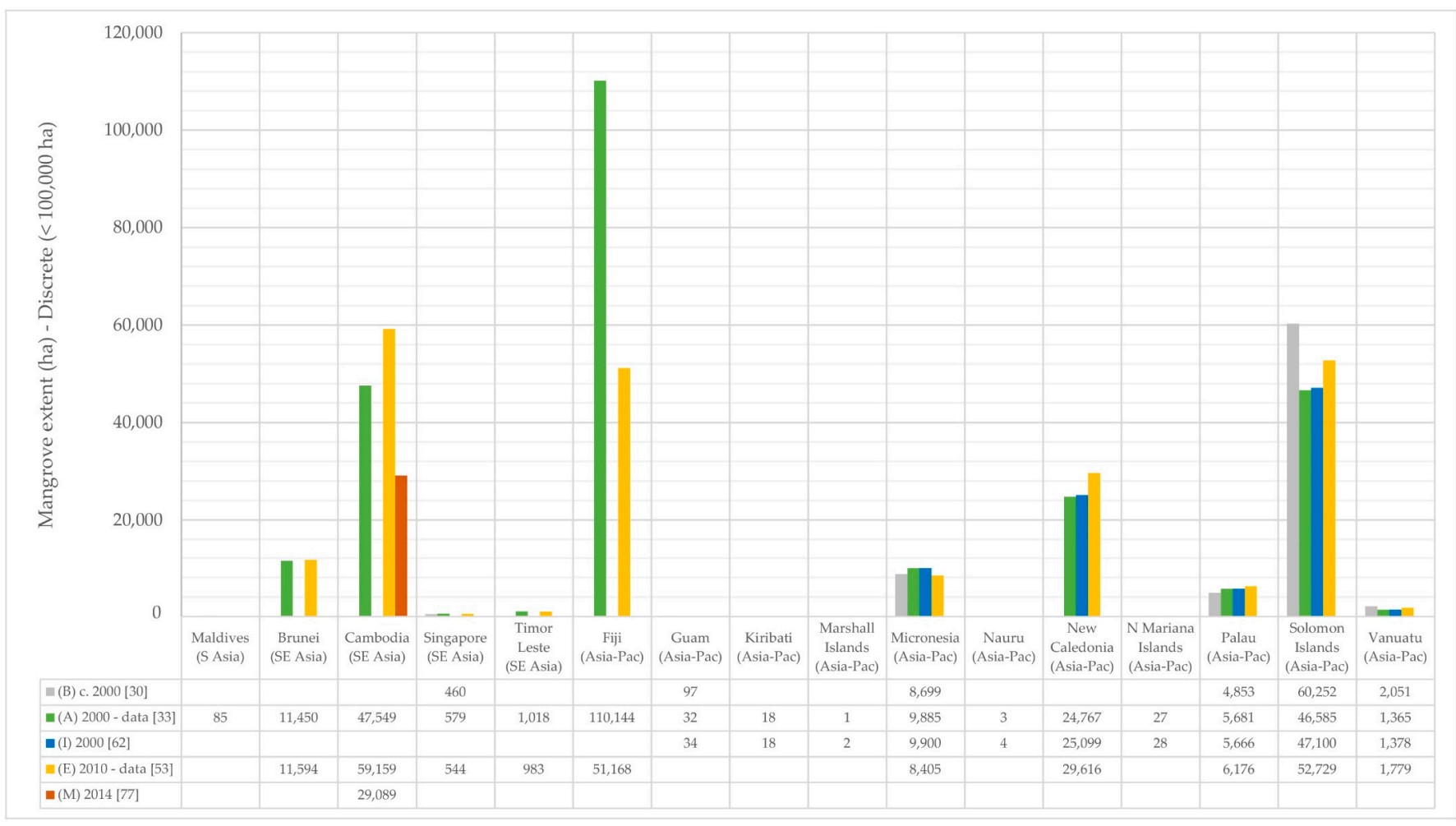

(**b**)

**Figure 2.** (**a**) Mangrove distribution (>100,000 ha) by country/territory, year and study—discrete datasets; (**b**) Mangrove distribution (<100,000 ha) by country/territory, year and study—discrete datasets.

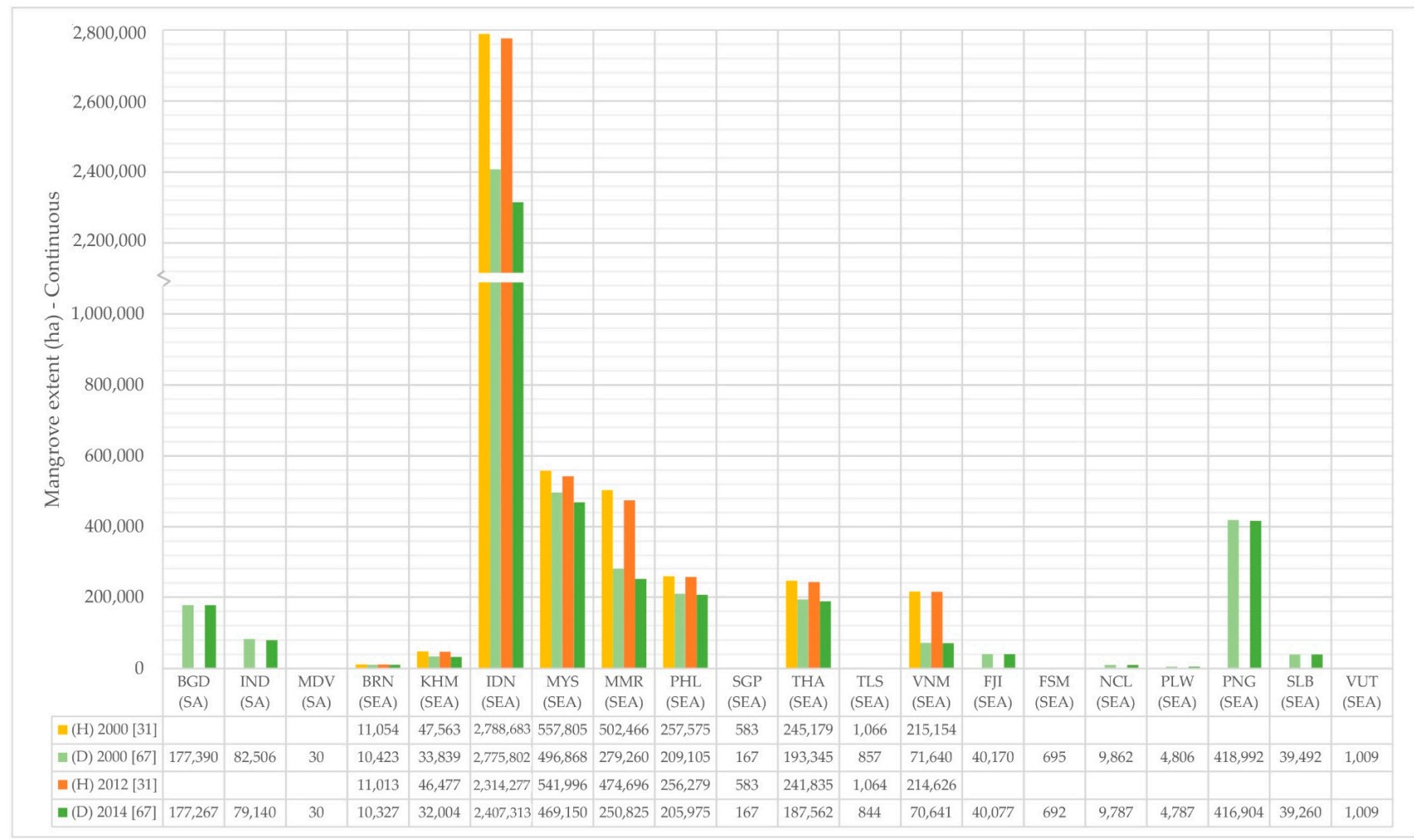

**Figure 3.** Mangrove extent (ha) by country/territory (using ISO Alpha-3 codes), year and study—continuous datasets.

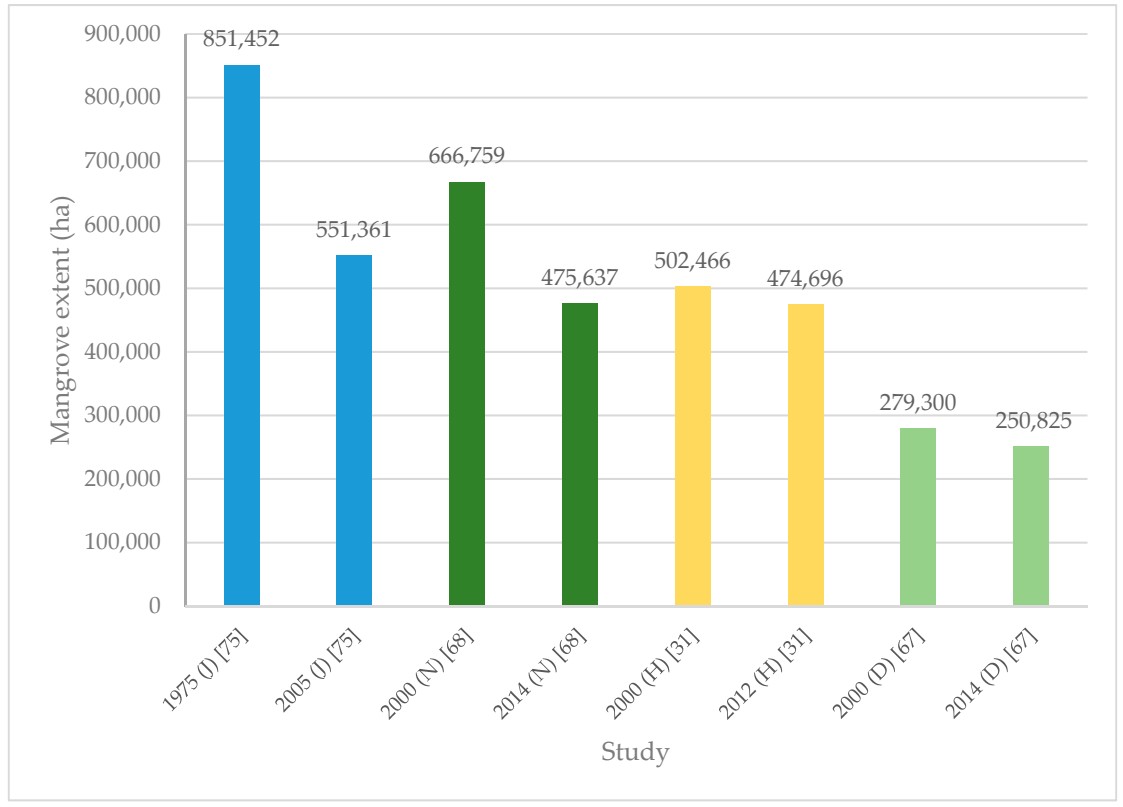

(**a**) Myanmar

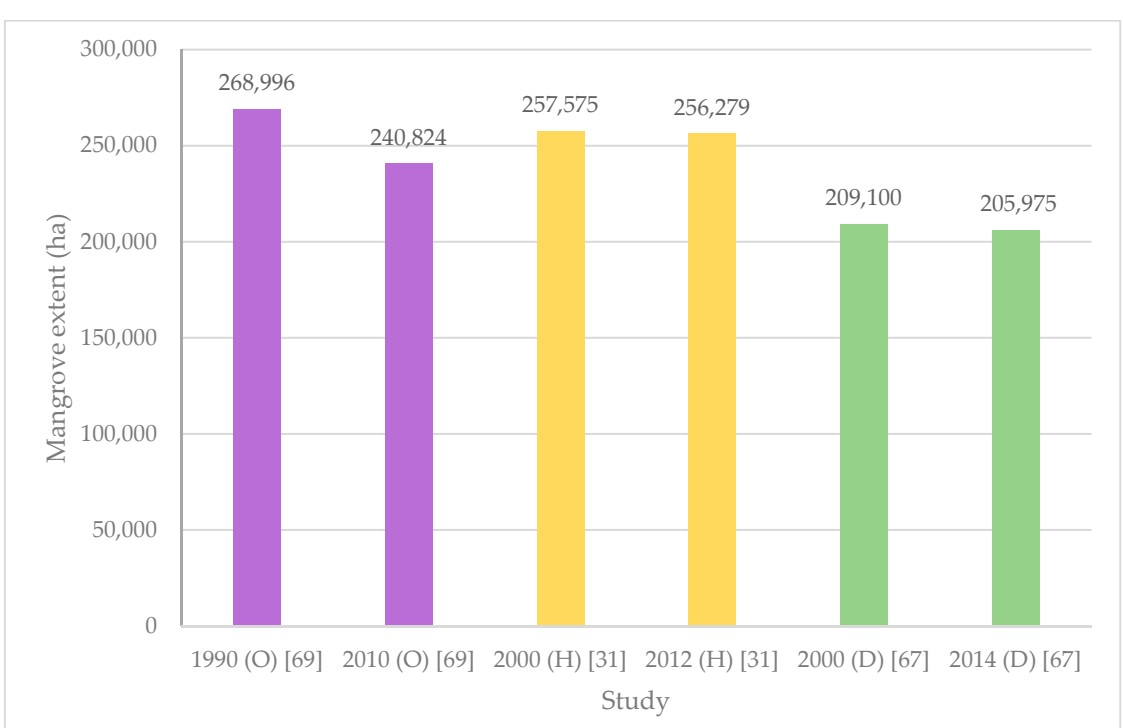

(**b**) Philippines

**Figure 4.** Mangrove distribution for primary loss hotspot countries ((**a**) Myanmar and (**b**) Philippines) by year and study.

**Table 5.** Mangrove extent (ha) by country/territory, year and study—continuous datasets. No continuous data is available for Guam, Kiribati, Marshall Islands, Nauru and Northern Mariana Islands, therefore these are excluded from this table.

| | | Study, Year of Focus, Extracted from | | | |
|---|---|---|---|---|---|
| | | (H) Richards and Friess [31] | (D) Hamilton and Casey [67] | (H) Richards and Friess [31] | (D) Hamilton and Casey [67] |
| | | 2000 | 2000 | 2012 | 2014 |
| Country | Sub-region | Paper | Supporting Info | Paper | Supporting Info |
| Bangladesh | S Asia | | 177,390 | | 177,267 |
| India | S Asia | | 82,506 | | 79,140 |
| Maldives | S Asia | | 30 | | 30 |
| Brunei | SE Asia | 11,054 | 10,423 | 11,013 | 10,327 |
| Cambodia | SE Asia | 47,563 | 33,839 | 46,477 | 32,004 |
| Indonesia | SE Asia | 2,788,683 | 2,407,313 | 2,775,802 | 2,314,277 |
| Malaysia | SE Asia | 557,805 | 496,868 | 541,996 | 469,150 |
| Myanmar | SE Asia | 502,466 | 279,260 | 474,696 | 250,825 |
| Philippines | SE Asia | 257,575 | 209,105 | 256,279 | 205,975 |
| Singapore | SE Asia | 583 | 167 | 583 | 167 |
| Thailand | SE Asia | 245,179 | 193,345 | 241,835 | 187,562 |
| Timor Leste | SE Asia | 1066 | 857 | 1064 | 844 |
| Vietnam | SE Asia | 215,154 | 71,640 | 214,626 | 70,641 |
| Fiji | Asia-Pacific | | 40,170 | | 40,077 |
| Micronesia | Asia-Pacific | | 695 | | 692 |
| New Caledonia | Asia-Pacific | | 9862 | | 9787 |
| Palau | Asia-Pacific | | 4806 | | 4787 |
| Papua New Guinea | Asia-Pacific | | 418,992 | | 416,904 |
| Solomon Islands | Asia-Pacific | | 39,492 | | 39,260 |
| Vanuatu | Asia-Pacific | | 1009 | | 1009 |
| Total | | 4,627,128 | 4,477,769 | 4,564,371 | 4,310,725 |

## 4. Conclusions

This report inventoried, described and compared seven single-date and eight multi-date datasets for 22 countries and three territories in S Asia, SE Asia and Asia-Pacific sub-regions. For all datasets, major limitations and challenges include the use of numerous definitions of 'mangrove' contributing to varying results; the lack of standardised and robust mangrove mapping methods (full methods are often not reported); a lack of coverage over much of the ROI; and the limited accessibility to many pertinent datasets. Despite these limitations/challenges, this inventory provides an important overview of what analysis of remotely sensed data has been conducted and the status of existing single- and multi-date datasets for the ROI. While single-date datasets are useful for providing snapshot distributions, they do not shed light on dynamics. In contrast, the comparison of multi-date datasets characterises regional change and identifies Myanmar as the primary loss hotspot. The Philippines, Malaysia, Cambodia and Indonesia follow Myanmar on a short-list of countries exhibiting highest rates of proportional mangrove loss across all inventoried multi-date datasets within the ROI. Future work can build on these results by acquiring and collating all available national and sub-national multi-date datasets for short-listed countries, conducting QAAs, and detailed deforestation analysis using best available or, if required, newly produced datasets. New mapping and monitoring efforts should take advantage of emerging cloud-based analysis platforms, such as Google Earth Engine. Efforts must also be taken to ensure all datasets are made available through online repositories with standardised metadata including details on projections, methods and accuracy. These measures will facilitate comprehensive and up-to-date assessments of dynamics. Accurate, detailed and timely dynamics information will help target loss hotspots within short-listed countries and guide intervention activities through programs such as the Blue Ventures (BV) Blue Forests (BF) project, which aims to support and enhance coastal livelihoods and safeguard biodiversity through the community-led restoration, conservation and managed-use of mangrove ecosystems within priority areas of mangrove loss. The BF project incentivises mangrove

conservation through a range of community-led fisheries management, forestry, and alternative livelihood-based incentives, including harnessing international voluntary carbon markets to generate credits—and income—for communities from the avoided deforestation of carbon-rich mangrove forests. In Madagascar, the BF project is further augmented through its integrated People Health and Environment (PHE) design, a holistic approach integrating community health service delivery with marine conservation and coastal livelihood initiatives within mangrove-dependent communities. Alternative livelihood initiatives being developed within and adjacent to mangrove ecosystems as part of this model include community-based ranching of hatchery-reared sea cucumbers, and apiculture for honey production. Ongoing efforts to expand this initiative in mangrove ecosystems throughout Madagascar are working towards the establishment of the world's largest mangrove conservation 'blue carbon' projects, an approach that BV is now replicating in the ROI. Results of this report are instrumental in informing country selection and prioritisation for replication efforts within this region. This includes informing international climate finance institutions of priority countries for investments in further 'blue carbon' mangrove conservation initiatives.

**Author Contributions:** Conceptualisation, T.G.J.; Data curation, S.G.; Formal analysis, S.G.; Investigation, S.G.; Methodology, T.G.J.; Supervision, T.G.J.; Visualisation, S.G.; Writing—original draft, S.G.; Writing—review and editing, T.G.J.

**Acknowledgments:** This report was funded by Blue Ventures Conservation, the UK Government's International Climate Finance (ICF) and the Global Environmental Facility's Blue Forest project. The report was further supported by the Department of Forest Resources Management in the Faculty of Forestry at the University of British Columbia.

**Conflicts of Interest:** The authors declare no conflict of interest.

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
