# Peer review of "Identifying Mangrove Deforestation Hotspots in South Asia, Southeast Asia and Asia-Pacific"

_remotesensing, doi:10.3390/rs11060728_

Round 1

Reviewer 1 Report

In the manuscript “Identifying mangrove deforestation hotspots in South Asia, Southeast Asia and Asia-Pacific” authors aim to present regional status of mangrove vegetation in Asia utilizing the available data and the literature. The topic, scope is of relevance and interest to the scientific community at large and valuable for mangrove forest management for climatic and anthropogenic vulnerabilities.  The article is lacking the remote sensing analysis. However, it seems that the literature has been used some remote sensing techniques for estimation of mangrove vegetation.

The major comments for main sections are as follows:

Abstract

The introductory part of the abstract is lengthy (L 12 – 21) and the methods are lean.

Results & Discussion

The authors have given the broad narrative for the datasets (A-P). However, there is a gap for consistency of some information. For example, some datasets have given the type of remote sensing data used i.e. IKONOS (A), but some dataset (D, F) were lacking the information.

Table 3 represents discrepancies of few countries and the statistics for some of those countries present thereafter. For example, figures for Myanmar in L 22-27; India in L 28-44 … It is good to organize paragraphs as the same order of countries represent  in Table 1 i.e. India, Indonesia,  Malaysia, Myanmar etc.

It was good to identify mangrove hotspots and quantified the mangrove forest change. However, it needs to give the procedures for identifying mangrove loss hotspots in methods section.

Table 4 and 5 show the mangrove distribution within country /territory. However, no findings were summarized within region i.e. south, southeast Asia and Asia-Pacific. In order to give a sense for the title, summarize the results according to region.  

NB: A few minor edits were made through sticky notes on the pdf.

Author Response

Dear Reviewer - please see the attached. 

Reviewer 2 Report

This paper reviewed the literature of global and station level mangrove dataset and build an inventory of forest loss hotspot. The inventory should be a valuable contribution. However, there is a major concern:

The study does not really compare and evaluate the dataset of all the studies. The discrepancies in the accuracy of the different dataset may make the combination of these datasets and the forest loss hotspot identification from it not reliable. For the paper to be considered for publication, it needs the real evaluation (e.g. uncertainties and accuracy in each dataset) of the datasets. 

Author Response

(The authors gave the same response as above.)

Round 2

Reviewer 1 Report

Authors addressed the comments suggested and presentation of the results are clear now. The figures and table carification looks good. The authors used the term"report" to indicate the research in several places. This is not a report and it is a scientific paper and therefore it is good to use a proper term.

Author Response

Dear Reviewer, 

Thank you for taking the time to re-read and comment on our revised manuscript.   We are very pleased to here that you feel we have addressed your concerns and that the results are now clear.  We have also now taken the extra measure of replacing the word "report" with study throughout our manuscript, ensuring proper reflection that this is a scientific study.  

We thank you for your time and patience. 

With very best regards, 

Trevor Jones and Samir Gandhi

Reviewer 2 Report

My main concerns have been addressed.

Author Response

Dear Reviewer, 

Thank you for taking the time to re-read and comment on our revised manuscript.   We are very pleased to here that you feel we have addressed your concerns and that the results are now clear.  We have gone back through and carefully proof-read one last time to ensure that any lingering English language issues have been addressed. 

We thank you for your time and patience. 

With very best regards, 

Trevor Jones and Samir Gandhi